# Effect of Fiber Bundle Morphology on Fiber Dispersion for Long Fiber-Reinforced Thermoplastics

**DOI:** 10.3390/polym15132790

**Published:** 2023-06-23

**Authors:** Hector Sebastian Perez, Allen Jonathan Román, Abrahán Bechara Senior, Tim Osswald

**Affiliations:** Polymer Engineering Center (PEC), University of Wisconsin-Madison, 1513 University Ave, Madison, WI 53706, USA; ajroman@wisc.edu (A.J.R.); bechara@wisc.edu (A.B.S.)

**Keywords:** fiber-reinforced plastics, long fiber-reinforced plastics, fiber dispersion, sliding plate rheometer, Couette rheometer

## Abstract

Understanding the mechanics of fiber attrition during the extrusion process is highly important in predicting the strength of long fiber-reinforced thermoplastic composites. However, little work has been done to investigate the mechanics of fiber dispersion and its effects on fiber attrition. This study aims at investigating fiber dispersion in simple shear flows for long fiber-reinforced thermoplastic pellets. Depending on the fabrication process, fiber bundles display distinct levels of compaction within the pellets. Studies have shown that morphological differences can lead to differences in dispersion mechanics; therefore, using a Couette rheometer and a sliding plate rheometer, coated and pultruded pellets were subjected to simple shear deformation, and the amount of dispersion was quantified. Additionally, a new image-based analysis method is presented in this study to measure fiber dispersion for a multi-pellet-filled system. Results from the single-pellet dispersion study showed a small amount of correlation between the dimensionless morphological parameter and the dispersion measurement. Pultruded and coated pellets were both found to have similar dispersion rates in a multi-pellet system. However, pultruded pellets were found to have a higher dispersion value at all levels when compared with coated pellets in both dispersion studies.

## 1. Introduction

Long fiber-reinforced thermoplastics (LFTs) have experienced increasing demand in the automotive and construction industries over the past decade. This is because of their exceptional mechanical properties, potential for lightweight construction, and processing advantages [1]. While past research efforts have primarily focused on analyzing fiber orientation, concentration, and length, more work is needed to evaluate fiber dispersion during plasticizing processing. Two main factors drive the study of fiber dispersion during the processing of LFTs. First, these materials are used to extrude different structural profiles for the construction industry, given their excellent impact and creep resistance in combination with their cost-effectiveness [1,2]. Excessive fiber damage is not typically a primary concern during the extrusion process due to the low level of stress. However, this low-stress level can result in a low level of fiber dispersion, which can pose problems in regions of the extruded components with low fiber content [3,4]. Therefore, maximizing fiber dispersion during extrusion becomes of great importance. Second, previous work has shown that undispersed fiber bundles can be a source of stress concentration in injection-molded parts [4,5]. This occurs as a fiber bundle produces dry surface areas with no adhesion to the polymer, weakening the part. Hence, optimizing dispersion while reducing fiber damage is vital for increasing the mechanical performance of components molded with LFTs.

LFTs are commonly supplied in a pellet form, where they can be manufactured with wire coating, crosshead extrusion, and pultrusion techniques. This study focuses on pellets manufactured with pultrusion and coating techniques. Pultruded LFT pellets are manufactured by pulling several rovings of fibers through an impregnation die where the rovings are impregnated with the polymer [6,7,8]. This technique often spreads the rovings with the dies, creating a greater fiber–matrix interface area. Coated LFT pellets are manufactured by pulling a single roving of fibers through an impregnation die. Parts produced with these types of pellets have been found to have a longer average fiber length compared with parts produced with pultruded pellets [7,8]. However, studies have also found that parts produced with coated pellets lead to fiber bundles within produced parts but also to a larger average fiber length [9]. Fiber bundles are a result of undispersed fiber bundles from the LFT plasticizing process.

Various studies have been carried out on the dispersion mechanics of carbon black agglomerates in simple shear flow [10,11,12,13]. While the shape and composition of E-glass fibers might be different from spherical agglomerates, the hydrodynamic effects that cause the dispersion process are similar and, therefore, can be used to explain fiber dispersion phenomena as well. Agglomerates have two methods of dispersion, erosion, and breakage [11,13,14]. Erosion is described as a slower dispersion process where individual particulates start separating from a larger bundle. Breakage occurs when a bundle breaks up into two or several bundles of particulates. These two methods of dispersion happen in a stochastic manner as a result of the internal composition of the fiber bundles within the pellet. Understanding these effects within a single pellet can be beneficial for a multi-pellet dispersion model, as executed by N. Domingues, 2010 [15]. The researcher was able to apply a rupture and erosion model to a flow field in a single-screw model. The effects and parameters that may cause a fiber bundle to erode or break up need to be studied. M. Kuroda, 2002, characterized pellets based on their average fiber-bundle area and found that initial fiber bundles with a larger cross-sectional area led to a lower level of dispersion [16].

This work analyzed pultruded and coated pellets for their dispersion behavior by subjecting them to a simple shear flow in a sliding plate rheometer and Couette rheometer. The sliding plate rheometer was used to study the dispersion of a single pellet under simple shear deformation. A single-pellet dispersion study allowed the effect of individual morphological characteristics on dispersion to be evaluated for each pellet. This set of experiments specifically investigated the effect of a fiber bundle’s perimeter and area on the initial dispersion mechanism. In this part of the work, dispersion was evaluated based on the overall displacement of the fibers. The Couette rheometer was used to study dispersion for a multi-pellet-filled system. Instead of analyzing individual pellets, the material was considered a homogenous mixture of LFT pellets. As that is often the case with the extrusion and injection molding process, this type of study allows the interaction between pellets to be considered. Moreover, the dispersion was evaluated on the homogeneity of the sheared samples from X-ray imaging; the results from both dispersion studies are presented in this work.

## 2. Materials

### 2.1. Materials

Both materials used in this study were 40 wt% glass–fiber-reinforced polypropylene. The pultruded pellets (Celstran PPGF40) provided by Celanese (Dallas, TX, USA) had a nominal length of 10 mm and an average diameter of 18 μm (Figure 1a). The coated pellets (STAMAX PPGF40) were provided by SABIC (Geleen, The Netherlands) and had a nominal length of 15 mm and an average diameter of 19 μm (Figure 1b).

### 2.2. Morphology Characterization

To characterize the material, pellets were scanned using X-ray microcomputed tomography (µCT) to determine the boundaries of the fiber bundles and the matrix. During the characterization scanning process, a voxel size of 37.85 µm/pixels was used, the voltage was set to 80 V, the current to 120 A, the integration time to 1000 ms, the gain to 8, and the number of projections was 2200. The voxel size was maintained constant throughout the scans at 37.85 µm/pixel. This was used in all the characterization scans to ensure pixel resolution was not a variable. A mount was manufactured to scan pellets vertically to ensure the pellet’s cross-section was scanned properly (Figure 2a). Once pellets were scanned, the fiber bundles were characterized by measuring the bundle’s outer perimeter and the fiber bundle’s area using ImageJ 1.53 (National Institute of Mental Health, Bethesda, MD, USA) (Figure 2b). The perimeter of the fiber bundle signifies the boundary layer between the fibers and the matrix. The area of the fiber bundle signifies the amount of fiber content spread by either sizing or air. Figure 3 compares coated and pultruded pellets as shown by µCT scans. 

To measure both the perimeter and the area, a color threshold was implemented to distinguish between the matrix and the fibers. To ensure that the designated threshold was adequate, 25 pellets were measured to calculate an average area. Once the area for all pellets was within a 10% margin of the average area, the threshold was approved. This was done as the fiber content was assumed constant for each pellet for each type of material, and variation in areas would indicate different fiber contents for each pellet, which should not be the case if they were manufactured using the same process. The perimeter was only measured in the outer layer of the bundle as inner voids could not be distinguished between air, matrix, or the sizing used by the provider. Because the pellets were of different sizes, a dimensionless parameter *S* was proposed and used to compare the smaller pultruded pellets to the larger coated pellets, where *S* is described by Equation (1): (1)$$S=\frac{Fiber\;Bundle\;Perimeter^{2}}{Fiber\;Bundle\;Area}$$

The *S* parameter is a modified circular shape descriptor often used to describe the shape of objects compared to a circular shape [17]. The parameter can also calculate a value for each type of shape regardless of size. A circle will always have an *S* parameter of 12.56 regardless of size; a square and an equilateral triangle of the same area will have *S* parameters of 16 and 20.78, respectively. The dimensionless parameter was calculated at 350 cross-sections along the length of the pellet, and 300 cross-sections were analyzed along the shorter pultruded pellets. A graph of the shape parameter normalized by the average shape parameter throughout the length of a coated pellet can be seen in Figure 4. After analyzing several pultruded and coated pellets, it was found that the dimensionless parameter at the midplane of a pellet usually aligned with the lengthwise average *S* parameter. As shown in Figure 5, where the average *S* parameter is plotted in comparison to the midplane *S* parameter, it can be seen that both values are relatively similar; this can be seen by the dashed line, which represents the point where the average *S* parameters of the midplane and length are equal. The midplane shape parameter was then used as the characterization parameter, as it allowed for more rapid characterization of LFT pellets. To obtain a population shape parameter for each type of pellet, 192 pellets were characterized for each type of pellet to determine the population’s average fiber-bundle area, average fiber-bundle width, and average shape parameter, as shown in Table 1.

## 3. Single Pellet Dispersion

### 3.1. Sample Preparation

Initial sample plaques were made from virgin matrix material, where characterized pellets were later implanted. The initial plaques of 50 mm × 125 mm × 2.16 mm were compression molded using a hot press heated up to 220 °C at a clamping force of 4.45 kN. Initial plaques were first made with the absence of pellets to avoid formation of porosity and to accurately place the pellets in the correct orientation during the compression-molding process. Once the plaques were manufactured, a die punch was used to make two cavities perpendicular to the direction of the flow at the center of the plaque, as shown in Figure 6. Cavities were made perpendicular to the flow direction to test the characterized cross-section with respect to the principal straining axis. Pellets were placed in the cavities and merged with the matrix material using the hot press. This allowed the pellet to be surrounded by matrix material while conserving the original shape and orientation of the pellet. To ensure the compression-molding process did not alter the characterization parameters of our samples, a scan was performed on a sample plaque after implanting the pellet using the compression-molding machine. The scans showed no significant alteration in the shape of the fiber bundle within the sample plaque as a result of the experimentation process. Figure 7 shows a fiber bundle during the characterization process (Figure 7a), and once implanted into the matrix plaque (Figure 7b). It can be appreciated that the shape of the fiber bundle does not significantly change during the sample preparation process. 

### 3.2. Sliding Plate Rheometer

To apply deformation to the sample plaques, a sliding plate rheometer (SPR) was employed for this experiment. The rheometer in use was based on the design by [18,19] (Figure 8). The SPR was heated by containing it within a convection oven, and the sliding plate was displaced by an Interlaken 3300 universal testing instrument. The rheometer had an effective surface area of 100 × 230 mm^2^. The gap thickness of the SPR was chosen to be 2 mm, as the width of the fiber bundles falls below that gap size. A total deformation of 50 mm was chosen, as this is the maximum amount of deformation that still allowed for two sample plaques to be tested simultaneously. The velocity of the sliding plate was maintained constant throughout the experiment at a velocity of 20 mm/s, which equates to a shear rate of 10 s^−1^.

The experimental procedure used to implant and shear the samples properly was based on the experiments conducted by S. A. Simon [18]. Since the materials being tested had different thermal and rheological properties, different processing temperatures had to be used for each material to ensure comparable viscosities were reached. Rheological data were observed, and processing temperatures were used where both materials would have a viscosity of 576 Pa∙s. After, the samples were placed inside the sliding plate rheometer, ensuring that the straight edge of the plaques was colinear with those of the sliding plate. Once the sample plaques were inserted and secured between the moving and stationary plates, the plaques were given 15 min to reach an isothermal state at the desired temperature. Once the desired temperature was reached, the plates were tightened to a gap of 2 mm. Since the gap of the SPR was smaller than that of the initial sample plaque, the sample was slightly compressed to ensure complete contact with the plates. Upon imposing the deformation at the specific deformation rate, the SPR’s heat was turned off, and the sample was then removed once the temperature had reached room temperature. Afterwards, samples were cut into rectangles 25 mm × 75 mm to isolate the fibers of each pellet.

### 3.3. Single Pellet Dispersion Measurement

Samples were scanned using the µCT Scanner to determine the position of the fibers after deformation. Scans were done using the same voxel size for each sample to ensure constant pixel resolution throughout the analysis process. During the scanning process, a voxel size of 40.96 µm/pixels was used, the voltage was set to 80 V, the current to 100 A, the integration time to 3000 ms, the gain was set to 4, and the number of projections was 2200. Figure 9 shows the fixture used to ensure the samples were vertical and separate from each other for every scan.

The scans were then converted into lengthwise stack images to measure the fiber content along the direction of deformation. A color threshold was used to differentiate between fibers and matrix material. The number of pixels characterized as fiber glass was measured at every stack image. The voxel size then allowed the separation distance between each stack image to be known, which then allowed for a fiber pixel weight distribution throughout each sample’s length, as shown in Figure 10. This allowed the range, concentration, and quantity of fibers to be measured along the length of the samples.

A histogram was constructed to depict the number of fiber pixels found throughout the length of the scan and to allow the dispersion measurements to be performed. When measuring the dispersion of a data set, a common measurement used is standard deviation [20]. Standard deviation was used to measure the spread of the fibers’ position, where a significant standard deviation represented a large spread of fibers in the sample. To measure the standard deviation of the fibers’ positions, Equations (2) and (3) were used. Equation (2) measures the mean position of the fibers, where $N_{s}$ is the total number of stack images, v is the voxel size that determines the distance between stack images, and $N_{f}$ is the number of fiber pixels found in each stack image. Equation (3) calculates the standard deviation of the fiber’s position.
(2)$$\bar{x}=\frac{\sum_{i=1}^{N_{s}}i\cdot v\cdot N_{f}}{\sum_{i=1}^{N_{s}}N_{f}}$$
(3)$$\sigma =\sqrt{\frac{\sum_{i=1}^{N_{s}}{\left(i\cdot v-\bar{x}\right)}^{2}\cdot N_{f}}{\sum_{i=1}^{N_{s}}N_{f}}}$$

## 4. Multi-Pellet Dispersion

### 4.1. Couette Rheometer

Fiber dispersion was studied for a multi-pellet-filled system by subjecting the fiber-reinforced pellets to a simple shear flow using a Couette rheometer. The experimental setup was developed by the Polymer Engineering Center at the University of Wisconsin-Madison, as seen in Figure 11 [21,22]. The device depicted in Figure 11 is composed of two concentric cylinders with an annular gap where pellets were placed in an arbitrary manner. The material was heated using a heating band wrapped around the device while a thermocouple was placed at the lengthwise mid-point to communicate with the heater controller. The outer cylinder is fixed while the inner cylinder rotates, creating a shear flow within the annular gap. The inner cylinder and heating band were controlled using a Brabender CWB/7.5 hp. The pellets were placed inside the gap between the cylinders with an arbitrary orientation. Once the material had melted, it was secured by a combination of brass rings and two threaded lids. Table 2 describes the experimental parameters used for this study. Once a deformation was applied, the material was cooled to room temperature. The sample was then extracted using a hydraulic car jack to push the sample out of the outer cylinder. Samples were cut in half using a heated blade, then heated to 165 °C for 15 min and flattened using a press. Four 35 mm squares were extracted in each experiment.

### 4.2. Multi-Pellet Dispersion Measurement

Samples were scanned using X-ray tomography to measure dispersion. This technique allows the red–blue–green (RGB) images to be converted into a gray-scale image, allowing the analysis of the pixel intensity values. After gray-scale conversion, each image was represented as a matrix of values ranging between 0 and 255, where the value relates to the level of signal attenuation caused by the material which is being penetrated. The higher the density, the more attenuation was present, causing the pixel intensity value to increase. If a material was less dense in comparison to the rest of the composite, the pixel intensity value was smaller. The pixel intensity values allowed determination of homogeneity, where a fully dispersed material would have identical pixel intensity values for every pixel. In order to calculate the value of dispersion, an initial undispersed sample and a fully dispersed sample had to be created as reference points. The undispersed sample was prepared using a hot press to melt pellets into 50 mm × 125 mm × 4 mm plaques. The fully dispersed sample was made by subjecting the pellets to 180 s of shear flow at the same velocity and temperature as the experimental samples. Figure 12 shows the progression in homogeneity of samples according to their residence time.

The standard deviation of the pixel intensity values was then calculated for the samples and reference samples, where a low standard deviation represented a more homogenous material. As shown in Figure 12, some samples tended to have a lower concentration of material in some areas as a result of sample extraction or irregularities during sample preparation. To normalize the dispersion value, with respect to homogeneity only, the coefficient of variation (*COV*) was used to determine the dispersion percentage for each sample. The dispersion was calculated using Equation (4) by comparing each sample with the undispersed and fully dispersed samples, where *ε* is the dispersion percentage:(4)$$\varepsilon =\frac{COV_{sample}-COV_{0\%}}{COV_{100\%}-COV_{0\%}}$$

To ensure that the scanning process did not affect the dispersion values, all reference samples and the experimental samples for one set of process conditions were scanned simultaneously. This ensured that any irregularities from the scanning process would be applied equally to the reference and experimental samples for that given set of process conditions. Figure 13 shows one of the scans used for this study.

## 5. Results and Discussion

### 5.1. Single-Pellet Dispersion Study

Observations from the deformed samples showed a combination of the erosion and rupture breakup mechanisms, as seen in Figure 9. Dispersion for the single-pellet samples was evaluated based on the fibers’ separation distance. In order to quantify dispersion, the standard deviation was used to measure the dispersal of fibers on a sample. The results of the standard deviation were plotted in comparison with the *S* parameter. A comparison between the coated and pultruded pellets can be seen in Figure 14, below. All but two pultruded pellets were found to have a higher dispersion value, in terms of standard deviation, than all the coated pellets despite having similar *S* values. The trendlines also show that the *S* parameter had a more significant effect on the pultruded pellets than on the coated pellets.

It should be noted that the experimental procedure employed in this study did not measure all possible morphological factors that could affect the dispersion rate. One of these parameters is the anisotropy of a bundle’s shape. The anisotropy of a fiber bundle’s cross-section could have a significant effect as fiber bundles aligned in a perpendicular direction, with respect to the direction of the deformation, would have a more considerable amount of separation than those aligned in a parallel direction. Gopalkrishnan [23] and Boyle [12] characterized the morphology of agglomerates using a measurement of filler concentration. However, when this value was calculated for both types of pellets it was found to be very similar; therefore, it was not considered. Results show that the *S* parameter can gauge which pellets might lead to a larger dispersion rate based on the perimeter and area of a fiber bundle; however, the *S* parameter was unable to gauge this same dispersion rate when comparing two different types of LFT pellets.

### 5.2. Multi-Pellet Dispersion Study

Couette device experiments were performed under identical conditions for both types of materials in order to determine the effect of pellets’ morphology. As mentioned in the previous section, two reference samples were made to determine the level of dispersion for each experimental sample. An undispersed and a fully dispersed sample were made for each type of pellet. In order to compare the dispersion mechanisms of both materials, a single dispersion scale was used to calculate the dispersion value of each sample. The coated pellets were used as the reference samples in the global analysis, since, compared to them, the pultruded pellets showed an initial degree of dispersion prior to shearing. The dispersion results using this dispersion scale can be seen in Figure 15. 

In Figure 15, the pultruded pellets show a higher level of dispersion at every time interval. The results from the multi-pellet study make sense, as the initial state of the pultruded pellets was at a higher level of dispersion than that of coated pellets, as seen in Figure 11. This higher initial level of dispersion gave the pultruded pellets a head start in reaching a fully dispersed state. This observation was also reported by Kuroda [16], where it was shown that fiber bundles with an area larger than 2 mm^2^ tended to persist in the dispersion process much more than pellets with a lower area. This was also the case for the current results, as the coated pellets had an average area of 2.32 mm^2^ whereas the pultruded pellets had an average area of 1.71 mm^2^. 

While the pultruded pellets reached a dispersion value of 90% at 15 s, the dispersion rate stabilized and did not increase much more after those 15 s. This is important when fiber breakage is considered, as higher residence times after 15 s do not significantly increase the dispersion values and could lead to a lower overall fiber length. Since the coated pellets also had a larger initial fiber length, it could be a possibility that by the 30 s time interval the coated pellets had a larger average fiber length than the pultruded pellets as their fibers were not exposed to an additional residence time. This analysis would also eliminate the problem encountered by Gupta [3,4] where a higher initial average fiber length led to a less dispersed system. Knowing the residence time at which both materials reach the same level of dispersion could lead to a fair comparison of fiber length for both types of materials given the same process conditions. 

Measuring fiber dispersion with this method also showed the same exponential behavior as carbon black agglomerates [11]. Figure 16 shows the trend lines described by [11] using Equation (5), where $\dot{\gamma }$ is the shear rate, *t* is the residence time, and *k* is a fitting constant. However, as the pultruded pellet showed a higher initial level of dispersion, a time increment term was added in the form of $t_{0}$ to depict this initial dispersion state, as shown by Equation (6):(5)$$\varepsilon =1-e^{-k\cdot \dot{\gamma }\cdot t}$$
(6)$$\varepsilon =1-e^{-k\cdot \dot{\gamma }\cdot \left(t+t_{0}\right)}$$

## 6. Conclusions

The dispersion mechanism for a single fiber-induced polypropylene pellet was studied. Pellets were characterized by the morphological properties of the fiber bundles. A single-pellet dispersion study was carried out to identify the effect of a bundle’s perimeter and size on its initial dispersion mechanism. Results showed that the proposed shape parameter had a small amount of correlation between the dimensionless morphological parameter and the dispersion measurement. The shape parameter had no correlation when comparing pultruded and coated pellets, as both materials showed distinct dispersion values even when pellets had the same *S* parameter value.

A multi-pellet system was also used to study the fibers’ dispersion mechanics. A new method for measuring dispersion is proposed, where a non-dispersed sample and a fully dispersed sample are used as a reference scale. Samples were scanned using X-ray imaging and measured for homogeneity based on the pixel intensity from the scanned image. Results demonstrate that pultruded pellets achieved a higher value of dispersion more rapidly than coated pellets. However, coated pellets reached a similar level of dispersion at the 30 s time interval, at which point the dispersion rate of the pultruded pellets also appeared to level off. It was furthermore shown that fiber dispersion followed a similar exponential dispersion behavior to that of carbon black agglomerates. Future work will focus on implementing the new dispersion analysis method to study dispersion phenomena during the extrusion process. 

## Figures and Tables

**Figure 1 polymers-15-02790-f001:**
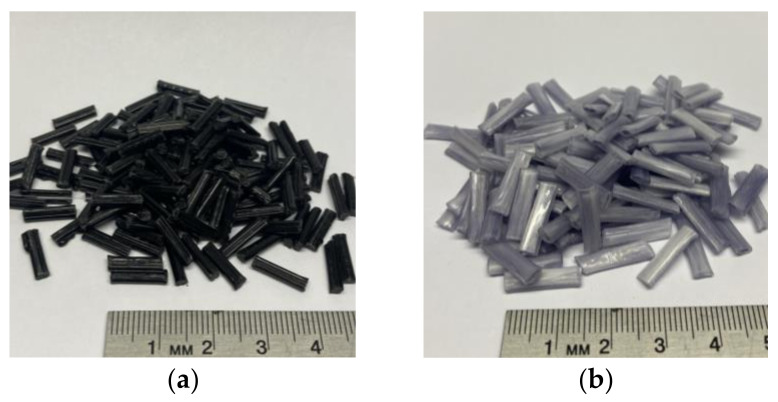
(**a**) Pultruded pellets provided by Celanese and (**b**) coated pellets provided by SABIC.

**Figure 2 polymers-15-02790-f002:**
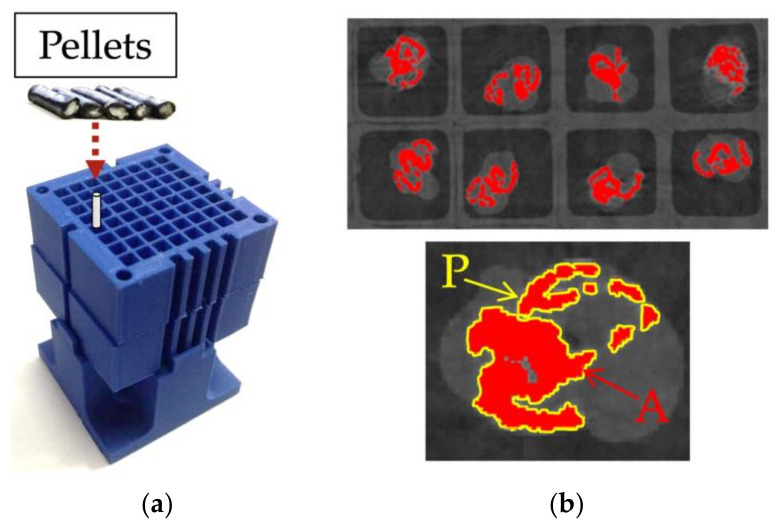
(**a**) Pellets were inserted vertically in a 3D printed mount; (**b**) cross-section of a scan where perimeter (P) and area (A) were calculated for a fiber bundle in a pultruded pellet.

**Figure 3 polymers-15-02790-f003:**
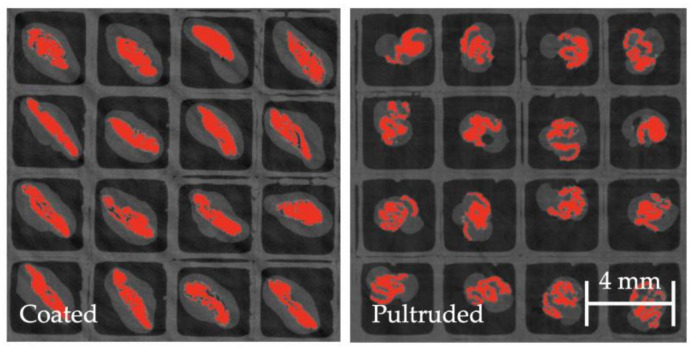
µCT scans of coated and pultruded pellets used for this study. Both images use same length scale.

**Figure 4 polymers-15-02790-f004:**
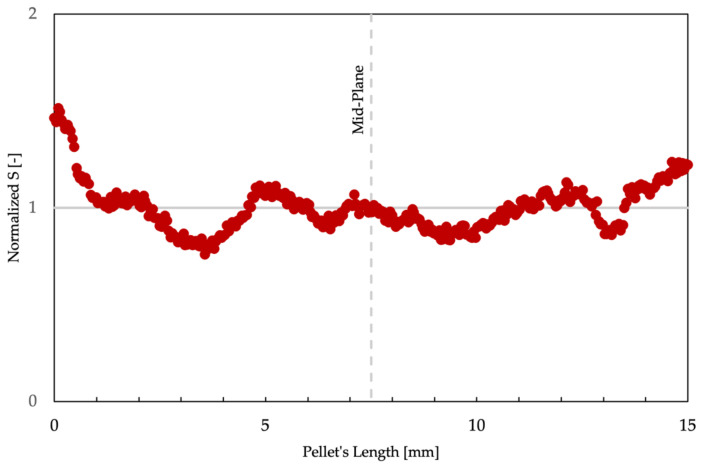
Normalized *S* parameter throughout a coated pellet’s length for a coated pellet.

**Figure 5 polymers-15-02790-f005:**
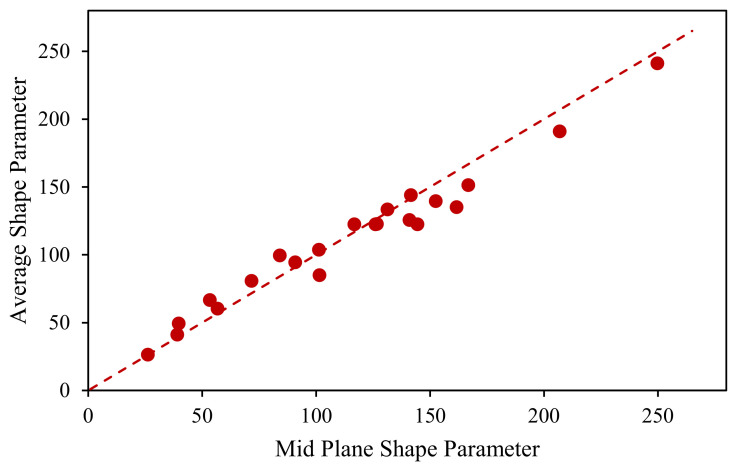
Average shape parameter of a pellet versus the mid-plane shape parameter.

**Figure 6 polymers-15-02790-f006:**
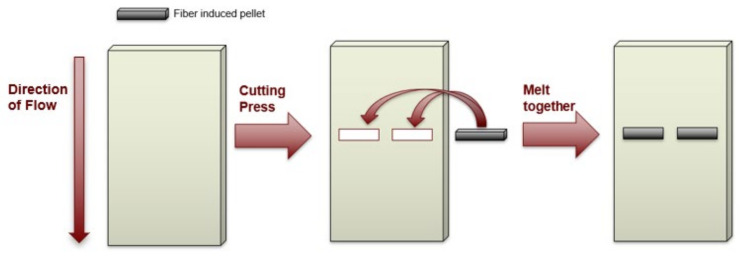
Procedure for imbedding LFT pellets into matrix sample plaque.

**Figure 7 polymers-15-02790-f007:**
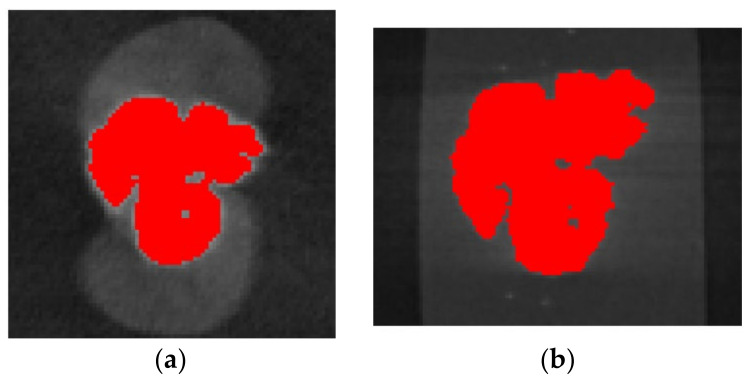
(**a**) Pellets were inserted vertically in a 3D printed mount; (**b**) cross-section of a scan where a perimeter and area were calculated for a fiber bundle in a pultruded pellet.

**Figure 8 polymers-15-02790-f008:**
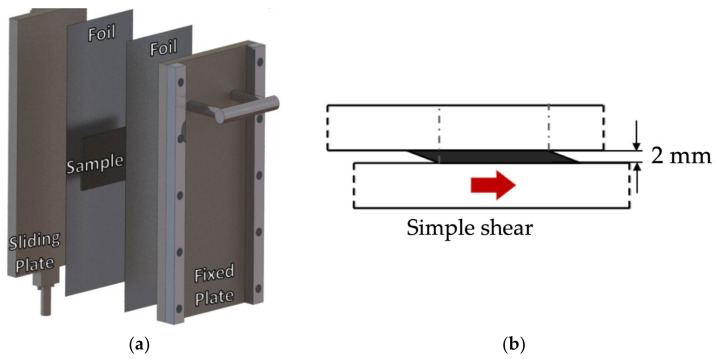
(**a**) Layers of sliding plate rheometer; (**b**) Side view of sliding plate rheometer with the sample after deformation.

**Figure 9 polymers-15-02790-f009:**
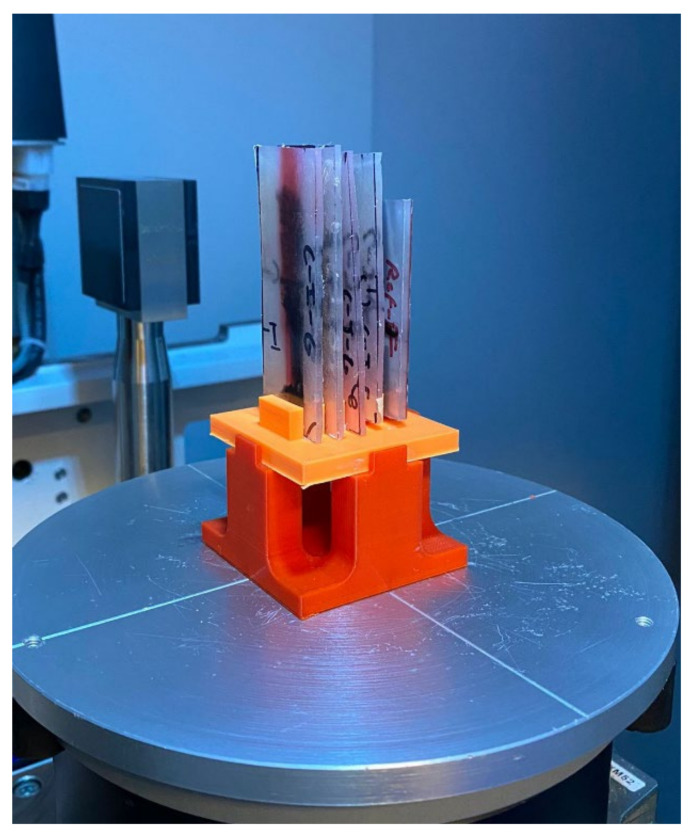
Mount used to scan samples with constant parameters.

**Figure 10 polymers-15-02790-f010:**
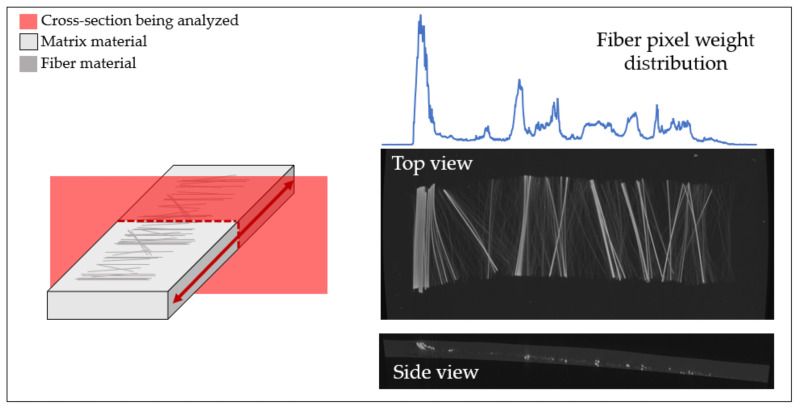
Fiber pixel weight distribution throughout the length of the sheared sample.

**Figure 11 polymers-15-02790-f011:**
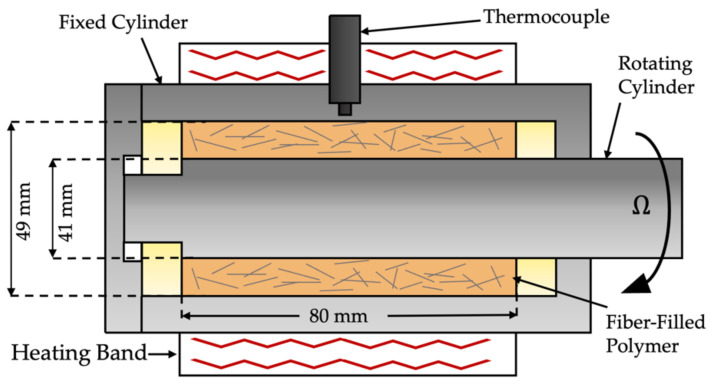
Schematic of Couette rheometer experimental setup.

**Figure 12 polymers-15-02790-f012:**
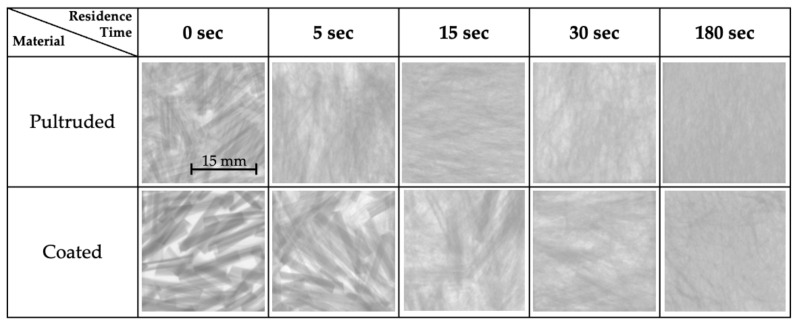
X-ray scans of samples for different residence times depicting the dispersion progression. All images use same length scale.

**Figure 13 polymers-15-02790-f013:**
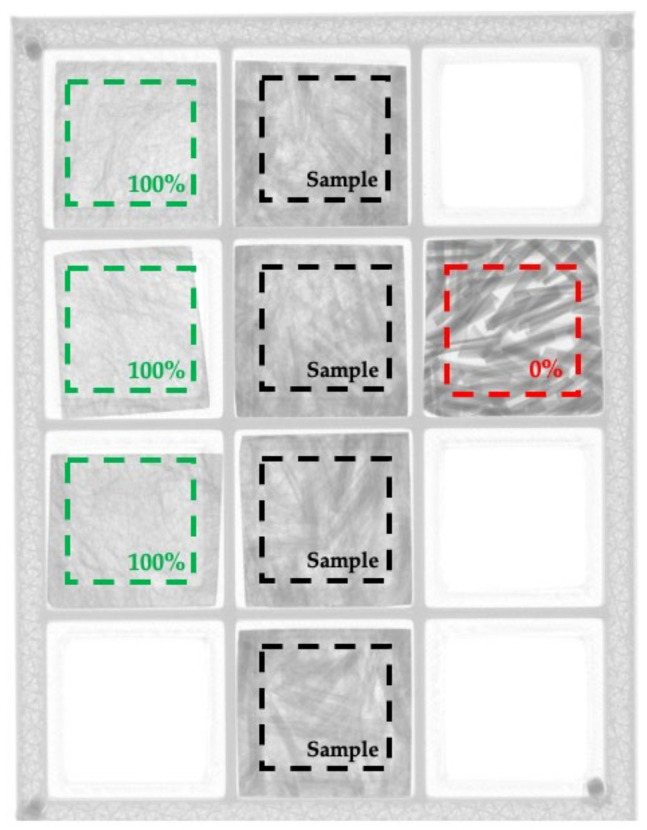
X-ray scan of samples and reference samples.

**Figure 14 polymers-15-02790-f014:**
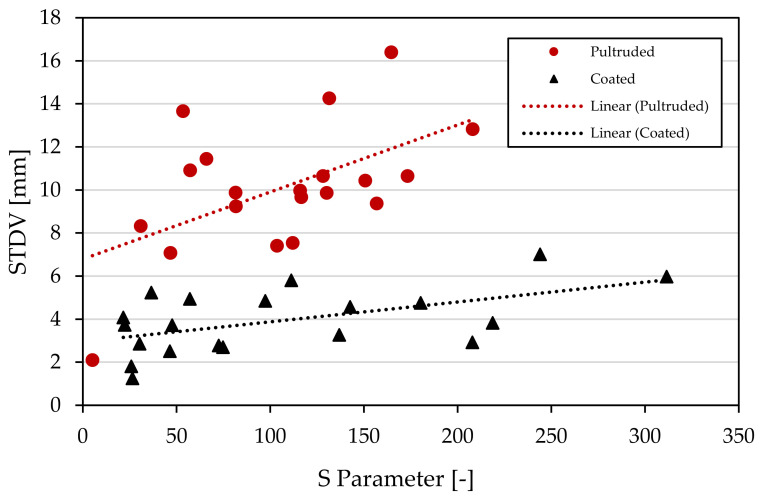
Standard deviation of the fibers’ spread versus the *S* parameter calculated from the characterization process.

**Figure 15 polymers-15-02790-f015:**
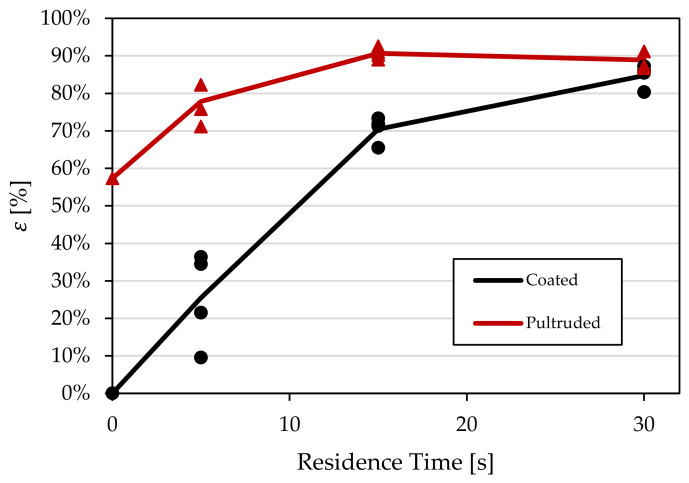
Dispersion results from the multi-pellet study.

**Figure 16 polymers-15-02790-f016:**
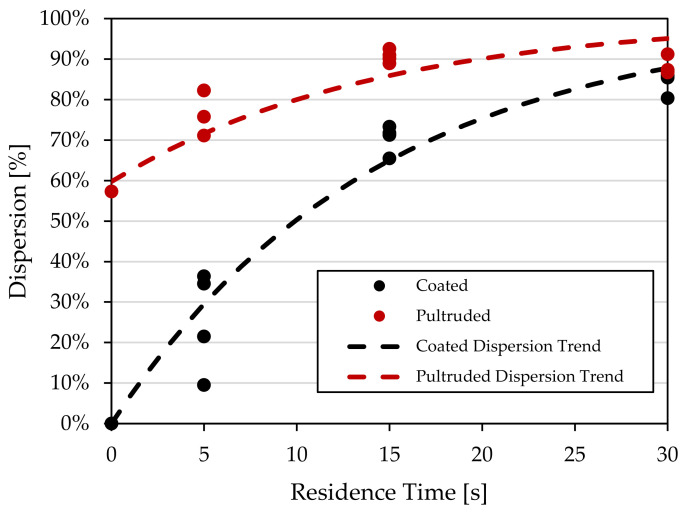
Dispersion trendline as an exponential function of total deformation with dispersion measurements (fitting constants *k* = 0.007, $t_{0}$ = 0 for coated trend, $t_{0}$ = 13 for pultruded trend).

**Table 1 polymers-15-02790-t001:** Morphological properties of sample set and population set.

Metric	Coated Pellets	Pultruded Pellets
Average Fiber Bundle Area [mm^2^]	2.32	1.71
Average Fiber Bundle Perimeter [mm]	9.24	13.03
*S* Parameter Average [-]	37.98	103.83

**Table 2 polymers-15-02790-t002:** Experimental parameters for multi-pellet dispersion study.

Metric	Levels
Fiber weight [%]	40
Melt viscosity [Pa]	501
Shear rate [s^−1^]	10.19
Residence time [s]	5, 15, 30, 300

## Data Availability

The data that support the findings of this study are available from the corresponding author upon reasonable request.

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
