# Peer review of "Effect of Fiber Bundle Morphology on Fiber Dispersion for Long Fiber-Reinforced Thermoplastics"

_polymers, 2023, doi:10.3390/polym15132790_

Round 1
Reviewer 1 Report
The Authors present a well-structured and organized paper that takes up a very interesting and significant issue related to understanding the mechanics of fibre attrition during the extrusion process. This issue is crucial in the context of plastic processing.
The writing style is clear and understandable while maintaining scientific rigour.
Comments for consideration:
- It would be interesting to add photos of pellets (coated and pultruded),
- Fig.1 - consider adding lines from P to the perimeter line and A to the area. At first reading, it is not clear that the colour of the font refers to the perimeter/area.
Minor comments:
- lack of spaces:
line 125, before "[17]",
lines 177 and 194, before "mm",
line 151, after 125 (and the unit is missing),
- lines 151 and 263, different styles of dimensions (it could be unified),
- par. from line 172, why are units bolded?
Have you carried out or are you considering conducting tests on an extruder?
Author Response
Thank you for your comments and suggestions. Below you will find a list with the modifications based on your comments. The updated manuscript is also attached.
- A picture of each type of pellet was added to the materials section in the form of Figure 1
- Arrows were added to Figure 1 (now Figure 2) to distinguish where the perimeter and area were calculated.
- Spaces were added where needed, as well as consistent styles of dimensions.
- Yes, we have already conducted some dispersion experiments in an extruder which will be another paper we will try and publish soon. I added a sentence in the conclusion section stating this is part of future plans.

Reviewer 2 Report
The manuscript presents the effect of fiber bundle morphology on fiber dispersion by comparing a pultruded and a coated long fiber reinforced thermoplastic. The paper is scientifically sound and well written.
In line 152 metric units for the clamping force at least in brackets would be appreciated.
The sentence in line 153/154 is hard to understand and should be rephrased.
The authors should explain briefly, why the LFT pellets were placed perpendicular to the direction of flow in the single pellet experiments.
For clarity, the authors should mention that only LFT pellets were added into the Couette rheometer as only the fiber weight percentage is given in table 2 and hence, only with the information of the materials section it can be deducted that solely LFT pellets were used in the multi pellet experiments.
Is the fiber orientation in figure 10 representative for the fiber orientation in the samples or are the fibers in the schematic only depicted in such way that it is visible, where the LFT is in the rheometer?
Very minor mistakes can be eliminated during proof reading.
Author Response
Thank you for your comments and suggestions. Below you will find a list with the modifications based on your comments. The updated manuscript is also attached.
- Line 152 (now Line 153) depicts metric units
- Sentence from line 153-154 (now Lines 154-156) is rephrased for clarity
- An explanation for the placement orientation of the pellets is given in the sentence from Lines 158-159.
- It is now stated in several sentences (Lines 234-235, 237-238, 243-244) that only pellets are placed within the couette rheometer for the experiments.
- It is now stated that Figure 10 (Now figure 11) depicts an arbitrary fiber orientation (Lines 237-238)
- Gramatical modifications have also been done throughout the entirety of the paper.
